# Relative Distribution of DnaA and DNA in *Escherichia coli* Cells as a Factor of Their Phenotypic Variability

**DOI:** 10.3390/ijms26020464

**Published:** 2025-01-08

**Authors:** Sharanya K. Namboodiri, Alexander Aranovich, Uzi Hadad, Levi A. Gheber, Mario Feingold, Itzhak Fishov

**Affiliations:** 1Department of Physics, Ben Gurion University of the Negev, Beer-Sheva 8410501, Israel; namboodi@post.bgu.ac.il (S.K.N.); aaranov@bgu.ac.il (A.A.); mario@bgu.ac.il (M.F.); 2Ilse Katz Institute for Nanoscale Science & Technology, Ben Gurion University of the Negev, Beer-Sheva 8410501, Israel; uzihad@bgu.ac.il; 3Department of Biotechnology Engineering, Ben Gurion University of the Negev, Beer-Sheva 8410501, Israel; glevi@bgu.ac.il; 4Department of Life Sciences, Ben Gurion University of the Negev, Beer-Sheva 8410501, Israel

**Keywords:** phenotypic variability, *Escherichia coli*, DnaA, HU, intracellular distribution, cell content variability, protein mobility

## Abstract

Phenotypic variability in isogenic bacterial populations is a remarkable feature that helps them cope with external stresses, yet it is incompletely understood. This variability can stem from gene expression noise and/or the unequal partitioning of low-copy-number freely diffusing proteins during cell division. Some high-copy-number components are transiently associated with almost immobile large assemblies (hyperstructures) and may be unequally distributed, contributing to bacterial phenotypic variability. We focus on the nucleoid hyperstructure containing numerous DNA-associated proteins, including the replication initiator DnaA. Previously, we found an increasing asynchrony in the nucleoid segregation dynamics in growing *E. coli* cell lineages and suggested that variable replication initiation timing may be the main cause of this phenomenon. Here, we support this hypothesis revealing that DnaA/DNA variability represents a key factor leading to the enhanced asynchrony in *E. coli*. We followed the intra- and intercellular distribution of fluorescently tagged DnaA and histone-like HU chromosomally encoded under their native promoters. The diffusion rate of DnaA is low, corresponding to a diffusion-binding mode of mobility, but still one order faster than that of HU. The intracellular distribution of DnaA concentration is homogeneous in contrast to the significant asymmetry in the distribution of HU to the cell halves, leading to the unequal DNA content of nucleoids and DnaA/DNA ratios in future daughter compartments. Accordingly, the intercellular variabilities in HU concentration (CV = 26%) and DnaA/DNA ratio (CV = 18%) are high. The variable DnaA/DNA may cause a variable replication initiation time (initiation noise). Asynchronous initiation at different replication origins may, in turn, be the mechanism leading to the observed asymmetric intracellular DNA distribution. Our findings indicate that the feature determining the variability of the initiation time in *E. coli* is the DnaA/DNA ratio, rather than each of them separately. We provide a likely mechanism for the ‘loss of segregation synchrony’ phenomenon.

## 1. Introduction

Bacterial cells display high phenotypic flexibility in response to environmental changes. Species that grow and multiply via symmetrical cell division (e.g., *E. coli*) eventually generate (presumably) identical daughters. Accordingly, in an isogenic population, all subsequent generations would be expected to display identical phenotypes in an unvarying and homogeneous environment. Experimentally, varying phenotypes appear even under these conditions. Phenotypic variability in isogenic bacterial populations helps them cope with external stresses [1,2,3], including exposure to antibiotics [4]. A detailed analysis of individual cells in a population can reveal and quantify this variability. The current understanding attributes gene expression noise and unequal protein partitioning during cell division the key role in the emergence of this variability [5,6,7]. This is essentially correct in the case of low-frequency gene expression, or that of low-copy-number freely diffusing proteins randomly shared between daughter cells. However, it ignores some high-copy-number components that are permanently or transiently associated with large, mostly immobile, assemblies, called hyperstructures [8], and thereby are not freely diffusing. Our hypothesis is that the hyperstructure organization of bacterial cells contributes to bacterial phenotypic variability.

In previous work, we studied the nucleoid hyperstructure that contains numerous DNA-associated proteins [9]. We found an increasing asynchrony of nucleoid segregation in a cell lineage after 2–3 divisions from a single cell. In contrast, the segregation synchrony remained high in filamentous lineages, wherein the lack of septa allowed the free diffusion of proteins along the filament. Nucleoid partitioning is the consequence of two processes: the ongoing DNA replication and the segregation of newly synthesized DNA strands. The segregation process is timed and, most likely, driven by ongoing chromosome replication [10,11,12,13]. The segregation timing is therefore determined mainly by the initiation time of chromosome replication and the segregation rate in a cell. Since we found the same segregation rates of nucleoids in dividing and filamentous lineages, the nucleoid segregation asynchrony in different cells implies that the replication initiation time is the main source of variability for the segregation process [9]. The initiation time is set by the interplay between the accumulation of the active form of the replication initiator DnaA and the cell growth [14], whereas a large fraction of DnaA is bound to the nucleoid.

DnaA has approximately three hundred high- and low-affinity binding sites on the chromosome, called DnaA-boxes [15]. DnaA exists in ATP-bound (initiation-active) and ADP (inactive) forms, the interconversion between which is cell cycle-dependent [16]. According to the titration model, initiation begins when the cell reaches the ‘critical’ mass: the threshold concentration of free DnaA-ATP is achieved after DnaA boxes on the chromosome (comprising the specific datA locus) are saturated by de novo synthesized protein [14]. In addition, the combination of activation and deactivation regulatory systems (through hydrolysis and exchange of nucleotides) may generate stable oscillations in the free DnaA-ATP concentration during the cell cycle [17].

This work examines whether the variability in the number of cell-inherited DnaA and its effective local concentration may serve as an essential source of variability in the replication initiation time and the ensuing chromosome segregation. In our previous works [9,18], the chromosomally encoded histone-like nucleoid-associated protein HU-eGFP was exploited to highlight the chromosome. The HU protein is one of the most abundant nucleoid-associated proteins with a low sequence specificity, playing multifaceted structural and regulatory roles in the bacterial cell (see references in [18,19] for a review). Here, in addition to HU-mCherry, we use chromosomally encoded DnaA fused with eGFP. This allowed for studying the intercellular and intracellular variability of the HU and DnaA content in the *E. coli* cell population. The mobility of both proteins is lower than that of freely diffusing proteins because they follow a transient-binding type of diffusion. However, the relatively higher diffusion rate of DnaA allows its equal distribution between daughter cells in contrast to the asymmetric segregation of DNA. This leads to a highly variable ratio of DnaA to cellular DNA amounts, and, presumably, to a significant variation in the replication initiation time.

## 2. Results

### 2.1. Spatial Distribution of DnaA-eGFP and HU-mCherry Within E. coli Cells

#### 2.1.1. Colocalization of HU and DnaA Is Notable in Living Cells but Changes When Transcription, Translation, or Replication Are Impaired

a. Fluorescence images of HU-mCherry and DnaA-eGFP overlap

In this work, we used DnaA with eGFP inserted within the domain II as described [20,21], and expressed from the chromosomal *dnaA* locus under the native promotor (see Section 4.1.1 and Appendix A). Together with the chromosomally encoded HU-mCherry, the fluorescence detected from both proteins reflects their real amounts and distributions within the cell. The double-tagged cells displayed a normal growth rate and size distribution (Appendix A), confirming the good functionality of the modified proteins.

The overall distribution of DnaA in the *E. coli* cells was examined firstly by comparing images in phase contrast with the GFP (for DnaA) and mCherry (for HU) fluorescence images and the corresponding intensity profiles (Figure 1 and Appendix A). Typically, the spatial distribution of DnaA fluorescence in untreated cells was wider than that of HU (Figure 1, upper row), roughly matching the nucleoid(s) contour. This also manifests in the corresponding fluorescence intensity profiles (Appendix A). However, the relatively large cell volume fraction occupied by nucleoids did not allow for reliable distinguishing between nucleoid-associated and free cytoplasmic DnaA. We therefore used various drug treatments to modify the relative volume fraction of nucleoids. The nucleoid appears highly compacted after the chloramphenicol (CAM) treatment [22]; however, surprisingly, the majority of DnaA fluorescence comes from the cytoplasm, even being essentially reduced in the nucleoid region of numerous cells (Figure 1, second row, Appendix A). This phenomenon may be due to the dissociation of the protein from DNA upon CAM treatment and calls for measuring the DnaA mobility to verify whether it is freely diffusing (see Section 2.2). Alternatively, the arrest of the DNA replication using nalidixic acid (NAL) [23] and the continued cell elongation leads to a remarkable increase in the cytoplasmic fraction, despite the nucleoid being somewhat stretched (Figure 1, third row). In this case, DnaA fluorescence fills the entire cytoplasm with a small fraction still concentrated in the nucleoid area. The additional CAM treatment of these cells led to moderate nucleoid compaction but did not change the distribution pattern of DnaA (Figure 1, fourth row). The inhibition of RNA synthesis with rifampicin (RIF) resulted in nucleoid expansion to the entire cell volume [24,25] and a similar spread out of the DnaA (Figure 1, last row).

b. Relative Spots positions

The co-localization analysis is relatively straightforward whenever proteins form clusters visible as distinct fluorescent spots. In our case, the HU-mCherry distribution and, to a larger extent, that of DnaA-eGFP are rather wide (Figure 1 and see profile plots in Appendix A). While the DnaA distribution is not completely homogeneous, we were not able to identify distinct ‘foci’ like those previously reported in *E. coli* [21,26] and *B. subtilis* [27]. For HU-mCherry, binding to chromosomes randomly and with low affinity, the highest local intensity merely highlights the maximal DNA density in the nucleoid. In contrast, DnaA has several high-affinity binding sites that may be visible as spots in a fluorescence image [26]. HU-mCherry and DnaA-eGFP fluorescence maxima (Figure 1, the ‘spots’ columns) may be regarded as spots and counted using an appropriate threshold (Section 4.2.1). The positions of spots relative to a pole are shown in Figure 2. In live, untreated cells (Figure 2A), HU spots reflect the number of nucleoids in cells of different ages: starting from two nucleoids in short cells and evolving into four nucleoids in the longer ones towards division. The distribution of the number of DnaA spots per cell is similar to that of the number of HU spots (Figure 2A, inset). DnaA spots display a remarkable position coincidence with HU spots, following the essential overlap of their intensity profiles (Figure 1, first row). This is confirmed by quantifying distances between related spots (Figure 2F, dark blue box, and Table 1). The distribution of distances is relatively narrow, with 2/3 of them being within the 0.5 µm range, well within the nucleoid body.

After the treatment of cells with CAM, the nucleoids are expectedly compacted and fused pairwise (Figure 2B, inset), while DnaA is spread across the rest of the cytoplasm (Figure 1, second row) and most cells lack the DnaA spots (Figure 2B, inset). The rare DnaA-eGFP spots are located ‘outside’ the zone of the HU-mCherry spots (Figure 2B), hinting that DnaA-eGFP detaches from the compacted nucleoid after CAM treatment. Accordingly, the distribution of distances becomes rather wide (Figure 2F, orange box) with only a quarter of spot pairs within a 0.5 µm distance (Table 1). In NAL-treated cells with a single nucleoid and a single DnaA spot, there is an overlap of spot positions as seen in live cells (Figure 2C), despite the remarkable spread of DnaA throughout the cytoplasm (Figure 1, third row). The distances between corresponding spots are large (Figure 2F, grey box), but still within the elongated nucleoid. However, when the cells are treated with NAL and afterward with CAM, the DnaA-eGFP spots are observed far from the HU-mCherry spots and outside the compacted nucleoids (Figure 2D,F, yellow box). In RIF-treated cells, both DnaA and HU appear to spread throughout the entire cell (Figure 1, bottom row) and, nevertheless, a single maximum of each protein can be detected as ‘spots’ using the same threshold as in Figure 2A–D (Figure 2E). The distribution of distances between paired HU and DnaA spots is narrower than in the case of both CAM and NAL-treated cells (Figure 2F, blue box), with about half of the distances being less than 0.5 µm (Table 1).

c. Pixel-wise colocalization

Yet another, more detailed approach to two fluorophores colocalization is the single-cell pixel-to-pixel analysis of their fluorescence intensities [28,29]. The pixel intensity values of the DnaA-eGFP and HU-mCherry images from the same cell are displayed as a scatterplot nicknamed cytofluorogram, which uses the respective intensities as coordinates (see Section 4.2.2). The distribution of the points reflects the extent to which the fluorophores are colocalized. While a positive linear correlation specifies a high degree of colocalization, a high density of points lying close (parallel) to one of the axes corresponds to the absence of colocalization. We also use the Pearson and Mander coefficients as quantitative measures of colocalization [29]. The Pearson coefficient provides the strength and mode of the correlation, while the Manders coefficient gives the degree of overlap between the two species.

The typical cytofluorograms of single cells in different functional states are presented in Figure 3. First, note the near-perfect colocalization of HU with DAPI (Figure 3F), confirming that HU-mCherry fluorescence provides an accurate representation of the DNA density: the highest intensities correspond to the nucleoid core, intermediate—the visible nucleoid body, and low—the nucleoid periphery. The cytofluorograms in Figure 3A–E display the colocalization of DnaA and HU. In a live cell (Figure 3A), many pixels with very low, close-to-background, fluorescence intensities (lower-left corner) correspond to fluorophore/protein-free cell areas. At higher intensities, a linear positive correlation is observed with a Pearson coefficient of about 0.8 (Appendix A), suggesting that more DnaA molecules reside at higher nucleoid densities, matching the close vicinity of the corresponding spots (Table 1). The cytofluorogram is rather different in CAM-treated cells (Figure 3B): most of the pixels within a wide range of DnaA-eGFP fluorescence intensities have a very low HU-mCherry fluorescence (laying parallel to the *y*-axis), indicating that they are located outside the nucleoid. A small fraction of high DnaA-eGFP intensities coincides with a wide range of mCherry intensities—within the nucleoid core. The Pearson coefficient value of 0.6 confirms a relatively poor correlation (Appendix A). NAL-treated cells have a slightly higher Pearson coefficient than CAM-treated cells (Appendix A), again with most of the GFP fluorescence localized elsewhere than the mCherry fluorescence-containing pixels (Figure 3C). Just part of the fluorophores colocalize according to the positively correlated intensities. Moreover, when the NAL-treated cells are additionally treated with CAM, the Pearson coefficient drops (Appendix A), and the distribution of the data points in the cytofluorogram (Figure 3D) is specific for the absence of colocalization. In RIF-treated cells, the homogeneous spread of both the nucleoid and DnaA-eGFP over the cell volume (e.g., Figure 1, bottom row) results in an almost perfect colocalization with a high Pearson coefficient value (Figure 3E and Appendix A). This, however, is not necessarily an indication of mutually dependent colocalization, since the same cytofluorogram will be obtained for two independent fluorophores homogeneously filling the same volume.

The Manders coefficient is remarkably high for the fraction of HU overlapping DnaA for all functional states. In contrast, the fraction of DnaA, which overlaps HU, is moderately high in live (untreated) and RIF-treated cells and decreases in cells treated with CAM (consistent with the images of Figure 1 and the patterns of the cytofluorograms). That means part of DnaA is always associated with the nucleoid, but a large fraction of DnaA resides in the nucleoid periphery in CAM-treated cells. Note that, in NAL-treated cells, the Manders coefficient is cell length-dependent, reflecting the lower nucleoid/cytoplasm ratio in longer cells.

These results suggest the colocalization of DnaA with the nucleoid in untreated cells and partially in NAL-treated, in contrast to a significant separation of the protein from the nucleoid occurring in CAM-treated cells.

#### 2.1.2. Distribution of HU and DnaA Between Cell Halves—Asymmetry Index

To reveal any asymmetry in the distribution of different components in the two cell halves and correlations between them, both randomly oriented and ordered according to the ‘leader pole’ cell populations were analyzed (see Section 4.3). We define ratios of the amounts/concentrations of these components in the two cell halves as asymmetry indexes. In a random population, the asymmetry indexes for HU, DnaA, and DNA amounts weakly correlate with the volume asymmetry with a wide variation between different cells (Figure 4A). As expected, the ratios of concentrations of HU-mCherry are well correlated with those of DAPI-stained DNA (Figure 4B), showing that HU provides a good representation of the DNA distribution in the cell [18]. In contrast, the concentration asymmetry of DnaA is only weakly correlated with that of HU, while indexes for volume and HU concentration anti-correlate (Figure 4B). In a randomly oriented cell population, the average asymmetry index of any variable is understandably close to 1 (Figure 5, Table 2, and see Section 4.3), but the cell-to-cell variability in the population is remarkable (Figure 4), with a CV up to 25% for HU and DAPI (Table 2).

Another way to assess the correlation in asymmetry between cell components is to evaluate it in a population aligned according to the cell half containing a larger amount of one of the components (‘leader pole’, see Section 4.3). Alignments corresponding to either a larger diameter or larger HU, DnaA, and DAPI amounts result in respective asymmetry indexes significantly higher than 1 (Figure 5A and Section 4.3). The HU, DnaA, and DAPI indexes do not follow the volume alignment contradicting the expectation that the asymmetry in the amounts of components with a homogeneous concentration throughout the cell will follow the volume asymmetry. When each of them is chosen as the leader pole for alignment, others do not necessarily follow. Accordingly, the concentration asymmetry indexes (Figure 5B) appear inversely asymmetric to the diameter-aligned cells. Notably, cells aligned according to amounts of HU or DAPI display high asymmetry indexes following each other (Figure 5A), consistent with the correlation found for the random population (Figure 4A). While this asymmetry in the distribution of DNA material between the cell halves also holds for the DNA concentration, it does not for the concentration of DnaA. Moreover, the DnaA concentration asymmetry index is close to unity for all types of alignments, meaning its concentration is homogeneous within cells. All concentration asymmetry indexes are summarized in Table 2. The difference between population-wide asymmetries of DnaA and DNA (DAPI or HU) concentrations is highly significant (*p* < 0.001) when cells are aligned according to the HU or DAPI content. Other asymmetry indexes do not differ significantly. We have also not found a remarkable dependence of asymmetry indexes on the cell size (age). Furthermore, the asymmetry in the distribution of DNA material between cell halves also holds for the wtDnaA MG1655 (*hu-egfp*) strain (Appendix A), confirming that it is not caused by a compromised functionality of DnaA-eGFP.

### 2.2. Mobility of HU and DnaA Is Impeded Within the Cell

To understand the difference in asymmetry in the intracellular distribution of HU and DnaA, the protein mobility was measured using FRAP (see Section 4.4). The diffusion coefficients of DnaA and HU were measured for live cells and cells treated with antibiotics. In addition, plasmid-borne eGFP and DNA-binding-impaired mutant DnaA(L417P)-eGFP expressed in the parent MG1655 strain were used as free diffusing controls.

In live cells (Figure 6), the mobility of DnaA appeared to be ten times faster than that of HU (Table 3), presumably reflecting their different affinities and number of binding sites on the chromosome. This diffusion coefficient for DnaA is very close to that obtained by FRAP and Single Molecule Tracking (SMT) [27]. The diffusion of HU was found much faster by SMT—0.14–0.38 µm^2^/s for the transition between binding sites [30,31]. The high number of binding sites and the residence time of proteins bound to DNA allow for estimating the apparent diffusion coefficient to be about 0.04 µm^2^/s [30]. This estimation is more relevant for comparison with the results of the FRAP method. Our rationale for using FRAP was to distinguish between the free (cytoplasmic) and the nucleoid-associated DnaA. It could not be achieved in live cells due to the large volume fraction occupied by the nucleoid, and the resulting high spatial overlap between the fluorescence of the two proteins (see Section 2.1.1. in (List a)). Therefore, bleaching merely the cytoplasmic DnaA was impossible, and only the ‘slow’, DNA-associated diffusion was detected.

Surprisingly, the diffusion of DnaA in the cytoplasm, outside the nucleoid that is either highly compacted by the CAM treatment or replication-arrested in elongated cells after NAL treatment, was even slower than in live cells (Figure 1, Table 3, Appendix A). The same slow diffusion of DnaA was also observed when the nucleoid was spread over the entire cell volume following RIF treatment (Table 3, Appendix A). The low fluorescence intensity of DnaA-eGFP, which limits the image acquisition rate, prevented us from detecting the fast fractions. As expected, GFP alone and mutant DnaA showed a one-order-faster diffusion rate than DnaA, typical for freely diffusing proteins lacking specific interactions with other structures [33].

### 2.3. The Binding Affinity of DnaA to the Nucleoid Is Weaker than That of HU

The presence of DnaA-eGFP fluorescence in the cytoplasm, outside the nucleoid, following CAM or NAL treatment (see Figure 1 and Appendix A), poses the question of the binding affinity of the protein to the nucleoid, assuming that the ‘cytoplasmic’ DnaA is free. Accordingly, we compared the DnaA and HU content in the nucleoid body of intact cells with nucleoids released from spheroplasts by measuring the nucleoid-associated fluorescence of DnaA-eGFP, HU-mCherry, and DAPI (see Section 4.1.3). The corresponding typical images (Figure 7) display nucleoids enclosed in the cellular envelope or released from bursting spheroplasts. DnaA fluorescence mostly overlaps with that of HU and DAPI in cells and spheroplasts but is essentially absent in the released nucleoids (Figure 7, lower-right image). Occasionally, some of the spheroplast ghosts collapse back and remain visible in phase contrast (blue arrows), trapping remarkable amounts of proteins, including DnaA (orange arrow) but not HU and DNA. Figure 8 provides a more detailed fluorescence image of nucleoids released from spheroplasts. Specifically, HU (B and C) and DAPI (C and D) exhibit different distributions within the nucleoid: while HU appears as a thread-textured cloud, DAPI staining reveals some additional compact structures that are absent in the HU-mCherry images. DnaA-eGFP is barely detected in the isolated nucleoid and is only visible within collapsed spheroplasts (A).

The loss of HU and DnaA from the nucleoid upon its release from spheroplasts is quantitatively presented as changes in the total HU-mCherry and DnaA-eGFP fluorescence intensities of a nucleoid relative to that of DAPI. This is in the assumption that the DAPI/DNA ratio is the same in different preparations due to its very high binding affinity [34]. Relative to the average amount of DnaA-eGFP per nucleoid in intact cells, a loss of 30% in the DnaA fluorescence is observed when the cells are converted to spheroplasts, while that of HU shows only a 6% drop (Figure 9). When the spheroplasts blow up, the DnaA fluorescence of the released nucleoids is reduced by 90%, and about 70% of HU is lost. Thus, the loss in the amount of HU in the isolated nucleoid was less severe, indicative of a stronger association of HU than of DnaA to the nucleoid. This qualitative estimation looks contradictive to the measured values of binding affinity for DnaA and HU to DNA: dissociation constants in a nanomolar range for DnaA [35] and a millimolar range for HU [32]. However, while there are about 300 such strong binding sites for DnaA on the chromosome of *E. coli* [15], HU can bind nonspecifically to 10^5^–10^6^ sites over the whole length of the chromosome so that the apparent dissociation constant of HU was estimated in a micromolar range [36]. This ‘sink’ effect results in a long retention time of HU on the nucleoid as a whole [36] and is consistent with the differences in the intracellular diffusion coefficients measured by FRAP (see Table 3).

### 2.4. Intercellular Variability in HU and DnaA Content in a Population

The finding of an asymmetrical distribution of the genetic material between cell halves (Section 2.1.2) raised a question about the variability of the DNA and DnaA cell contents in the population. We thus performed an analysis of the cell-to-cell variation in a steady-state culture. The linear correlation between total DnaA-eGFP fluorescence and the cell size (Figure 10A) and between DnaA and HU (Figure 10C) indicates the continuous synthesis of DnaA during the cell cycle proportionally to the DNA content. As a result, the concentration of DnaA is cell volume-independent (Figure 10D), i.e., it is constant during the cell cycle. This seems to be in contradiction with the single-time event of the initiation of DNA replication, meaning that initiation is not dependent, in general, on the overall cellular concentration of DnaA, but rather on the concentration of its activated free form. Yet, the DnaA concentration varies from cell to cell of the same volume (CV ~ 18%) (Figure 10D and Table 4). The HU cellular content is also proportional to the cell volume (Figure 10B), as was reported earlier [18]. However, its concentration shows a remarkably higher variability (CV ~ 26%) (Figure 10E and Table 4). Essentially, the methanol permeabilization used for DAPI staining (Section 4.1.2) did not affect the variability in the HU and DnaA.

The correlation between the concentrations of DnaA and HU is relatively weak with a covariance value of 0.73 (Figure 10F). This cloudy relationship may be better characterized by the variation in each of them, expressed as follows:(1)$$\eta_{DnaA}^{2}=\frac{\left\langle \left.{\left(d-h\right)}^{2}\right\rangle \right.\;}{2\left\langle \left.d\right\rangle \right.\left\langle \left.h\right\rangle \right.};\;\eta_{HU\;\;}^{2}=\frac{\left\langle \left.dh\right\rangle \right.-\left\langle \left.d\right\rangle \right.\left\langle \left.h\right\rangle \right.}{\left\langle \left.d\right\rangle \right.\left\langle \left.h\right\rangle \right.}$$
where *d* and *h* are the cell concentrations of DnaA and HU, respectively, and angled brackets denote the average over the population. The values of this variation, 0.024 for DnaA and 0.050 for HU calculated for the same population, support a much lower variability in DnaA concentration than that of HU. These features show the lack of synchrony in the variability in DnaA and HU.

The ratio of DnaA/HU is almost volume-independent (Figure 11), and its variance is notable (Table 4). Moreover, the variance is age-dependent—higher in smaller cells, decreasing towards division (Figure 11, orange line, Table 4). Notably, in a slow-growing culture, the variability in DnaA and HU concentrations, but not their ratio variability, are lower than in a fast-growing culture, and the difference between young and adult cells is less pronounced (Table 4 and Appendix A). These variabilities are much lower in cells treated with CAM or RIF (Appendix A, Appendix A). This might be the replication and division runout outcome, producing completed chromosomes segregated into smaller cells independently of the initially varying DNA content (see also the insets to Figure 2B,E).

## 3. Discussion

In live cells, the protein fluorescence was well associated with the nucleoid body highlighted by HU-mCherry (Figure 1). Although its distribution was somewhat heterogeneous, we did not observe distinct foci or helixes as reported for similar constructs in *E. coli* and *B. subtilis* [21,26,27]. This may be due to a different imaging setting optimized for collecting multiple sample fields to reveal the population variability (see Section 4.1.2).

Both the spot position and the pixel-to-pixel colocalization analyses (Section 2.1.1) indicate a significant overlap in the location of HU-mCherry and DnaA-eGFP in live cells, suggesting the tight association of DnaA with the nucleoid. This is supported by the low mobility of DnaA obtained by FRAP (Table 3): although it is one order faster than that of HU, it is one order slower than that of DnaA(L417P) or free GFP. This is consistent with the binding-diffusion mode of its motion reported by single-molecule tracking [27]. The wider-than-HU-mCherry fluorescence intensity profiles of DnaA-eGFP (Figure 1) and the moderate Manders overlap coefficient (Appendix A) imply that a part of DnaA resides in the nucleoid periphery or is unbound. The last option could not be examined due to the limited temporal resolution of FRAP at the long acquisition time (see Section 4.4). In fact, DnaA readily dissociates from the released nucleoids in contrast to HU (Section 2.3). The attempt to increase the nucleoid-free cytoplasmic volume by arresting protein or DNA synthesis brought an unexpected result (see below).

The HU content perfectly correlates with the cell size (Figure 10B), and with DAPI staining [18]; moreover, it follows the DAPI distribution within the cell (Figure 4 and Figure 5A), population-wide, representing the DNA content of nucleoids. This is obviously due to the huge number of HU binding sites over the entire chromosome length [38]. Within released nucleoids, the distribution of HU and DAPI looks somewhat different (Figure 8), presumably reflecting their different DNA topology-dependent binding modes [39]. The analysis of the cell asymmetry indexes (Figure 4 and Figure 5) indicates that DNA material is distributed unevenly within the cell, suggesting an asymmetric nucleoid segregation. On average, more DNA is found in the half of a larger volume (Figure 4A) consistent with that reported in [40] (Note that the volume is larger essentially due to the larger diameter since the cell length is equally divided in our calculation method). The correlation is weak and has a high variance, meaning that a larger nucleoid is not necessarily partitioned into the cell half of a larger volume. The DNA concentration thus negatively correlates with the volume (Figure 4B) and its asymmetry index in HU-aligned cells is about 1.15 (Table 2). The consequence of this intracellular asymmetry is the high intercellular variability of HU concentrations in a population (Figure 10E,F, Table 4). We also note the same CVs of asymmetry indexes and the cellular concentrations (Table 2 and Table 4).

DnaA does not necessarily follow the distribution of HU or DAPI and appears well equilibrated between the cell halves. The amount of DnaA also correlates with the cell volume (Figure 10A), and its concentration asymmetry of cell halves is close to 1, at any cell alignment (Table 2). So how does the intercellular variability in DnaA concentration develop? The different DnaA/DNA ratios in divided cells should be created due to the high asymmetry in the DNA concentration in the mother cell. Accordingly, the DnaA/DNA ratio varies in cells (Table 4), with a larger variance in younger cells than in adults (Figure 11 and Table 4). Comparing the variability in HU (CV = 26%), DnaA, and their ratio (both with CV = 18%) (Table 4) together with the appreciable correlation between DnaA and HU concentrations (covariance value of 0.73, Figure 10F), it can be concluded that HU and DnaA are not completely independent variables. Namely, a larger nucleoid can bring more bound DnaA to a newborn cell. The low fraction of free DnaA (total [41] minus bound [15]) may be considered as a low-copy-number noise factor. Thus, DnaA/DNA variability may lead to variable replication initiation times, creating initiation noise. At least a partial equilibration of DnaA between multiple nucleoids along filamentous cells may establish a higher synchrony of initiations, supporting the explanation of our previous findings [9].

The estimated initiation asynchrony index of 0.14 for fast-growing *E. coli* [42] is compatible with our 1.15 intracellular asymmetry of HU. Analyzing some discrepancies between the cell mass and chromosome content of cells treated with CAM or Rif in their pioneer work, Skarstad et al. noted that they reflect “the cell-to-cell variation in initiation and termination ages” [43,44]. In addition, fast oscillations of DnaA between cell halves with a period of about 5 s were detected in *E. coli* [27]. Such oscillations may occasionally initiate replication in one of the two nucleoids only. The fast inactivation of DnaA-ATP just after the initiation [16] would delay the initiation in the nucleoid in the other cell half. The nature of these DnaA oscillations is unknown, but their possible relation to the “longitudinal density waves” in replicating nucleoids in the same time scale [45] seems very intriguing. Could it be that DnaA is ‘relocated’ by these waves along or between nucleoids?

Asynchronous initiation at different oriCs may be, in turn, a plausible origin of the asymmetric DNA distribution within the cell. At our growth conditions, newborn cells contain two already-initiated nucleoids (with 4 oriCs) and four nucleoids in cells approaching division (Figure 2A). Two pairs of new replication forks in one nucleoid can accumulate a 15% asymmetry in the DNA content of nucleoids during a 3–4 min difference in initiation times. This period is well comparable with the average time difference (asynchrony) between nucleoid segregation events in cell lineages [9]. The intercellular HU concentration varies even more (CV = 26% (Table 4)), suggesting an additional uncertainty factor. The asymmetric cell division [40,46] is suggested to be such a factor. This type of asymmetry was not explored in this work. In addition, the high variability in the time of the start of septation (CV of ~30%, [47]) may also result from variabilities in replication initiation and nucleoid segregation, permitting the Z-ring formation.

In the slow-growing *E. coli* population, the variabilities in HU and DnaA are lower (Table 4), apparently because of the lower number of oriCs per cell (one or two), and the initiation and corresponding termination are within the same cycle. The resulting initiation asynchrony index measured in the same strain was 0.06—distinctly lower than 0.14 in fast-growing cells [42]. The fact that the DnaA/HU variance is not decreased to the same extent may indicate a significant division asymmetry, generating daughter cells with an equal amount of DNA but a higher amount of DnaA in the larger cell.

Attempting to discriminate between nucleoid-associated and freely diffusing DnaA, we exploited cells with different nucleoid/cytoplasm volume ratios: with nucleoid compacted after CAM treatment or expanded cytoplasm in elongated NAL-treated cells (Figure 1, second and third rows). As a result, DnaA appears to ‘spread’ outside the nucleoid (see the corresponding fluorescence profiles in Figure 1). Unexpectedly, the diffusion of DnaA in cytoplasmic areas of both kinds of cells was even slower than in untreated cells (Table 3, Appendix A). This slow diffusion may be attributed to the DnaA bound to the few high-affinity DnaA-boxes on the DNA strands outside the compacted nucleoid body. CAM inhibits ribosome translocation and stabilizes mRNAs [48], while the RNA synthesis persists [25,49]. Super-resolution microscopy reveals 46% of bound RNAP in the periphery of nucleoids compacted by CAM treatment [50]. Our visual impression of the compacted nucleoid boundaries may be misleading, while the HU and DAPI fluorescence intensity profiles display, in addition to the central narrow peak, noticeable shoulders covering the rest of the cell (Appendix A). This profile can be deconvolved into two Gaussian peaks—one high and narrow, corresponding to the bright nucleoid body, and another low but stretched over the cell length (Appendix A). Given the ten-times difference between the peaks’ amplitudes, the cell-wide ‘background’ is hardly identified by the eye, though comprising about 40% of the total cellular fluorescence intensity (comparable with the percent of the bound RNAP). This assumption regarding DNA distribution in CAM or NAL-treated cells and the corresponding DnaA binding requires further investigation beyond the aims of this study.

The major cell cycle event in bacteria—the initiation of chromosome replication, is believed to be controlled by two mechanisms: the accumulation of DnaA to the threshold level [14] and regulation of the DnaA activity (ATP/ADP forms) [16], or both [17]. The timing of this event and its relation to the cell size (the ‘initiation mass’) was a matter of interest for many decades [51,52]. Although the data presented here are insufficient to distinguish between free and bound DnaA, nor its ATP/ADP form, it is understandable that the time required to establish the conditions for initiation depends on the starting conditions of the cell, e.g., the cell size, DnaA content, growth rate, etc. In one of the more recent works, this question was examined by changing the cell size [53]. It was concluded that in *E. coli,* the cellular amount of DnaA, rather than its concentration, is decisive for the initiation timing. We now specify that the DnaA/DNA ratio may be a more rigorous feature in determining the variability in the initiation time in *E. coli*. We suggest the variability in the DnaA/DNA ratio as the main factor of the variable replication initiation time in *E. coli* (see Figure 1). This can explain the phenomenon of increasing asynchrony in the nucleoid segregation in a growing cell lineage reported in our previous work [9].

## 4. Materials and Methods

### 4.1. Strains and Sample Preparation

#### 4.1.1. Strains and Plasmids

The MG1655 strain of *E. coli* was used in this study carrying chromosomal copies of two genes in their original loci, *dnaA* and *hupA*, fused in frame with fluorescent reporters, *egfp* and *mCherry*, respectively. The construction of the strain is described in the Appendix A.

DnaA(L417P)-mCherry and eGFP proteins were expressed in the wild-type MG1655 transformed with pBAD24 plasmids carrying the corresponding genes [20] by the addition of 0.2% arabinose for 2.5 h.

Antibiotic treatment (where relevant) was performed at 37 °C with 100 µg/mL CAM, 180 µg/mL Rif for 60 min, and 30 µg/mL NAL for 30 min. Cells transformed with pBAD24 plasmids containing antibiotic resistance were grown with 50 µg/mL ampicillin. For FRAP experiments, the cells were treated with low Cephalexin (15 µg/mL) for one generation time.

#### 4.1.2. Sample Preparation for Microscopy, Image Acquisition, and Processing

*E. coli* MG1655 (*dnaA-egfp*, *hu-mcherry*) were grown exponentially for 8–10 generations in Neihardt’s medium [54] (basal plus supplement) (FORMEDIUM Ltd., Hunstanton, UK) at 37 °C to the optical density 0.2 at 600 nm. The generation time was 32 min in the full medium and 65 min without the supplement. For the microscopy, cells from steady-state cultures were loaded on Neihardt’s 2% agarose gel pads and imaged at room temperature. For the DAPI staining, the cells were washed twice with PBS by centrifugation at 16,000× *g* for 5 min at 4 °C and the final pellet was resuspended in ice-cold PBS containing 10% methanol. The permeabilized cells were stained with 1.5 µM DAPI and loaded on PBS 2% agarose pads.

Images were taken using a Zeiss Axiovert 200M fluorescence microscope equipped with a 100×/1.4 plan-apochromat objective, and an AxioCam HRm camera (all from ZEISS MI, Carl Zeiss, Oberkochen, Germany). The microscope operation and image acquisition were controlled by the AxioVision SE (V4.9.1.0) program. A reduced light intensity (1D neutral filter in the excitation light path) was used to minimize the photobleaching of fluorophores during multifield acquisition. Images of DnaA-eGFP and HU-mCherry were acquired with corresponding filter sets and exposure times of 5 s and 500 ms, respectively. The GFP, mCherry, and DAPI images were subjected to a “rolling ball” background subtraction routine of ImageJ [55] using the 80-pixel ball radius. To decrease the noise level in the DnaA-eGFP images, an additional background creation with a 1-pixel ball radius was performed. Images of approximately 500 cells were acquired, processed, and analyzed using the Coli-Inspector project of the ObjectJ plugin for ImageJ [37].

#### 4.1.3. Spheroplasts Preparation and Release of Nucleoids

To estimate the binding affinity of HU and DnaA to the nucleoid, MG1655 (*dnaA-egfp*, *hu-mcherry*) cells were converted into spheroplasts, from which nucleoids were released. All three preparations were imaged using fluorescence microscopy, and the extent of retained fluorescence intensities in the released nucleoids, relative to that in intact cells, served as a measure of the affinity. Bacteria were grown as described above. An amount of 3 µM DAPI was added for DNA staining. The lysozyme-induced spheroplasts were prepared as described in [56]. Briefly, 10 mL samples of cells from steady-state growing cultures were centrifuged (5 min, 16,000× *g*, room temperature), and pelleted cells were resuspended in cold 0.8 mL buffer A (0.01M Tris, pH8, 0.1M NaCl, 20% sucrose) and 0.2 mL buffer B (0.12M Tris, pH 7.7, 0.05M EDTA, 0.4 mg/mL lysozyme). In 20–30 min of incubation on ice, ~90% of the cells were converted to spheroplasts.

The nucleoids were released from the spheroplasts heated at 37 °C for 10 min, loaded on the agarose pads prepared using buffer A, and immobilized on the agarose surface (Figure 7 and Figure 8).

The fluorescence intensities of DnaA-eGFP and HU-mCherry of released nucleoids were calculated using the DAPI-stained images of nucleoids as a mask. The GFP/DAPI and mCherry/DAPI ratios of fluorescence intensities in cells, spheroplasts, and released nucleoids were normalized to the average intensity ratios of intact cells.

### 4.2. Intracellular Spatial Distribution of DnaA-eGFP and HU-mCherry

#### 4.2.1. Scatter Plot of Spot Positions

The localization of DnaA-eGFP and HU-mCherry was analyzed using the Coli-Inspector plugin for ImageJ [37]. “Spots”, identified as the fluorescence maxima positions for each fluorophore, were generated by setting the tolerance limit at about 80% of the intensity maximum. The spot number for HU-mCherry was verified empirically according to the expected number of nucleoids in the cell, either from the CCSim simulation [57] or experimentally found for the same strain and conditions [9]. At our conditions, this number is between 2 and 4 nucleoids per cell, well detected with appropriate tolerance (Figure 2A). For generating the spots of DnaA-eGFP we followed the approach of Nozaki et al. [26]. About 500 cells were analyzed in a population, and the histogram of HU spot numbers (i.e., number of nucleoids) correlates well with the cell size distribution (compare Figure 2, insets, and Appendix A). A similar distribution was observed for DnaA spots (Figure 2B). The spots’ positions for DnaA-eGFP and HU-mCherry relative to a cell pole were obtained and plotted to examine colocalization. DnaA and HU spots were also paired based on the closest proximity, and the corresponding distances between them were calculated.

#### 4.2.2. Cytofluorogram: Pixel Intensity-Based Correlation

In pixel-wise colocalization analyses, the intensity of a pixel in one channel is evaluated against the corresponding pixel in the second channel of a dual-color image, producing a scatterplot from which a correlation coefficient is determined. The ImageJ plugin JaCoP [28] (https://imagej.net/plugins/jacop version 2.1.1) was used to generate scatterplots and to characterize the degree of colocalization [29]. The cytofluorograms normalized to the intensities’ maxima were converted to color-coded density scatter plots using the MATLAB routine.

### 4.3. Population Analysis and Cell Asymmetry Measurement

The cellular amounts and concentrations of DnaA and HU were analyzed in a cell population by exploiting the single genome copies of *dnaA-egfp* and *hu-mcherry* (see Section 2.1.1). The total fluorescence intensity per cell served as a measure of the tagged protein amount in the cell since every DnaA and HU protein carries its fluorescent tag. The integral cell fluorescence in both GFP and mCherry channels, as well as the dimensions of corresponding cells, were obtained using the “Coli Inspector” program as described in [37]. Concentrations of DnaA and HU were calculated by dividing the total cell fluorescence intensity by the cell volume.

Cell diameter and fluorescence intensity profiles along each cell in a population are collected and stored in a “Map” image (Figure 12A). In these maps, each cell is shown as a line of the corresponding length in pixels, where the brightness of each pixel along the line corresponds numerically to the cell diameter (d, in pixel) or fluorescence intensity integral of a cell cross-section (slice) at this point. As each pixel-wide slice of a (rod-shaped) cell is close to a disk shape, its volume V_c_ can be calculated as V_c_ = π(d/2)^2^ cubic pixels. The whole cell volume (or its part) can be thus calculated more accurately as a sum of corresponding slices instead of using the mean cell diameter. Amounts and concentrations of DnaA-eGFP, HU-mCherry, and DAPI are also calculated more reliably using the volume calculated in this way.

**Figure 12 ijms-26-00464-f012:**
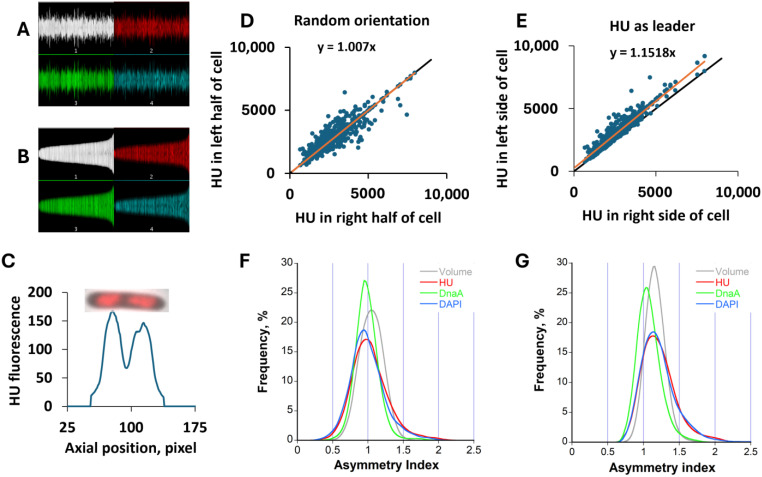
(**A**) Profiles maps of diameter (white), and fluorescence intensities of HU-mCherry (red), DnaA-eGFP (green), and DAPI (blue) in a population. (**B**) ‘Sorted’ map: same profiles as in (**A**) but sorted according to cell length. (**C**) Profile of the HU-mCherry fluorescence along an individual cell (above) with apparently more HU in the left half (the leader pole). (**D**,**E**) Relation between HU integral fluorescence in cell halves in randomly oriented (**D**) or aligned (**E**), HU as the leader pole populations; equality line (black) and the linear fit (orange) are shown. (**F**,**G**) Histograms of asymmetry indexes for volume (grey), HU (red), DnaA (green), and DAPI (blue) in random population (**F**) and aligned for each property (**G**).

The map can be sorted according to the cell length or other measured parameters (Figure 12B). Furthermore, the cells in a map can be aligned (‘flipped’ map) according to the wider or higher-intensity cell half (‘leader pole’). In this case, all cells in the map are oriented with the leader pole in the same direction, allowing for the analysis of intracellular distributions of different (labeled) components. This way, the ratios of volumes and integral fluorescence intensities between the cell halves (half length) were calculated for random populations or aligned using different leader poles. In a randomly oriented population, the ratios of all quantities in cell halves (asymmetry indexes) are distributed around 1 (Figure 12D,F). It was obtained occasionally slightly larger or smaller than 1, experimentally (Figure 5, Table 2, no leader pole), reflecting the one-pixel accuracy of the mid-cell demarcation. After alignment, the asymmetry indexes become larger than 1 (Figure 12E,G), revealing a single-cell asymmetry distribution (exemplified in Figure 12C) in the whole population.

The asymmetry index was thus calculated using the ratio of amounts or concentrations of HU, DnaA, and DNA in the two halves of each cell and averaged over the population. Alternatively, this index can be derived from the slope of the linear regression fit to the corresponding values in cell halves for the whole population (Figure 12D,E).

### 4.4. Protein Mobility Measurement Using FRAP

The Zeiss Axio-Observer 7 inverted microscope (ZEISS MI, Carl Zeiss, Oberkochen, Germany) was used with the 3i Marianas (Denver, CO, USA) spinning disk confocal microscope equipped with the Yokogawa W1 module (Yokogawa, Tokyo, Japan) and Prime 95B sCMOS camera (pixel size 0.106 microns) (Photometrics, Tucson, USA). The acquisition was controlled using SlideBook software (Intelligent Imaging Innovations, SlideBook v2304).

The pre-bleaching and recovery frames were acquired at a 100% illumination for both channels. The ROI area of 0.90 µm^2^ was exposed to 100% power of the 488 nm laser photobleaching GFP and mCherry. Two frame intervals were used—2 s and 500 ms for 3 min and 50 s recovery kinetics of HU-mCherry and DnaA-eGFP, respectively.

Fluorescence intensities of the chosen ROIs were measured after background subtraction using the 80-pixel rolling ball routine. The acquisitional bleaching of control (unbleached) cells was used for the data correction and normalization. The recovery half-time of DnaA and HU for each cell was derived from the single exponential fit generated by KaleidaGraph software (Synergy software, version 4.0). The predicted maximal recovery was calculated from the total cell and the bleached ROI areas. The diffusion coefficients of DnaA and HU were obtained using the simFRAP plugin version 0.95 [32] for Fiji software version 2.9.0 [58] based on simulating a 2-D random walk in each pixel of bleached ROI.

## Data Availability

The data presented in this study are available on reasonable request from the corresponding author.

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
