# Peer review of "Relative Distribution of DnaA and DNA in Escherichia coli Cells as a Factor of Their Phenotypic Variability"

_ijms, 2025, doi:10.3390/ijms26020464_

Round 1

Reviewer 1 Report

Comments and Suggestions for Authors

In this work, the authors investigate the cause of asynchrony in nucleoid segregation in an isogenic E. coli population. Based on the hypothesis that variable replication initiation timing is responsible for the asynchrony, the authors studied the distribution and mobility of DnaA and HU by imaging. Different phases of the cell cycle were represented by antibiotics treatment. From their data, they conclude that variable DnaA/DNA ratios in E. coli determine the variation of replication initiation time.

Overall, the manuscript is well written. The methods are thoroughly described. The data are concisely described and presented in adequate figures and tables. Figures are accompanied by appropriate captions. For each experiment, the rational, the setup, the data and the respective conclusions are comprehensively explained. In the end, the authors present a schematic, integrating their data in a model.

In summary, I recommend to accept the manuscript, following minor revisions.

Minor issues:

I like the strategy to express tagged proteins from the chromosome under the control of the endogenous promoter. How did you test that the expression level of the recombinant genes was indeed similar to the wild type expression level?

Page 18, lines 602-603: “To estimate the binding affinity of HU and DnaA to the nucleoid, we used to release nucleoids from cells and measure the extent of retained proteins relative to that in intact cells.”

This sentence needs to be rephrased.

Author Response

Comments and Suggestions for Authors

In this work, the authors investigate the cause of asynchrony in nucleoid segregation in an isogenic E. coli population. Based on the hypothesis that variable replication initiation timing is responsible for the asynchrony, the authors studied the distribution and mobility of DnaA and HU by imaging. Different phases of the cell cycle were represented by antibiotics treatment. From their data, they conclude that variable DnaA/DNA ratios in E. coli determine the variation of replication initiation time.

Overall, the manuscript is well written. The methods are thoroughly described. The data are concisely described and presented in adequate figures and tables. Figures are accompanied by appropriate captions. For each experiment, the rational, the setup, the data and the respective conclusions are comprehensively explained. In the end, the authors present a schematic, integrating their data in a model.

In summary, I recommend to accept the manuscript, following minor revisions.

Minor issues:

Comment 1: I like the strategy to express tagged proteins from the chromosome under the control of the endogenous promoter. How did you test that the expression level of the recombinant genes was indeed similar to the wild type expression level?

Response 1: We agree that examination of the expression levels of modified genes is important. In this work, we haven’t compared expression of the tagged proteins with their wild-type expression level. The functionality of the HU-FP fusions was tested in our previous works (Abebe et al., 2017; Gelber et al., 2021) as well as in other recent publications (e.g. Bettridge et al., 2021) and found unperturbed. Regarding the more important and essential protein DnaA, its expression was compared with the wild type in two labs, where this DnaA-FP construct was originally invented (Boeneman et al., 2009, Nozaki et al., 2009 – both cited in the manuscript). The expression level of DnaA-FP was found slightly lower than that of the wild type, but its initiation functionality appeared practically unaffected. In our construct, we essentially followed the strategy described in the original papers and, therefore, the same expression and functional properties are expected. Moreover, we have tested the functionality of cells containing genes of both tagged proteins and found the growth rate and size distribution identical to the wild type (Figure S1).

Comment 2: Page 18, lines 602-603: “To estimate the binding affinity of HU and DnaA to the nucleoid, we used to release nucleoids from cells and measure the extent of retained proteins relative to that in intact cells.”

This sentence needs to be rephrased.

Response 2: This sentence is now rephrased for clarity (lines 603-607): “To estimate the binding affinity of HU and DnaA to the nucleoid, MG1655 (dnaA-egfp, hu-mcherry) cells were converted into spheroplasts, from which nucleoids were released. All three preparations were imaged using fluorescence microscopy, and the extent of retained fluorescence intensities in the released nucleoids, relative to that in intact cells, served as a measure of the affinity.”

Reviewer 2 Report

Comments and Suggestions for Authors

Dear authors,

I appreciate the opportunity to review your interesting work phenotypic variability in E. coli due to DnaA and DNA distribution. Though the work justified the topic of interest, however, there are corrections and editing to be performed before making it to a publishable form. The results are explained well though seems to be very lengthy and hence needs to be clearer and more concise. Figures needs major attention specially the font size on axis and legends. My comments are summarized in the main manuscript for your review. 

Comments on the Quality of English Language

Proper proofreading of the English language is necessary, as there are many errors.

Author Response

Comment 1: The results are explained well though seems to be very lengthy and hence needs to be clearer and more concise.

Response 1: We are grateful for this reviewer’s thorough and constructive comments that helped us to improve the manuscript. Taking into consideration the various levels of experience and variety of research interests of the potential readers, we believe that a thorough (though lengthy) explanation has a higher priority than conciseness.

Comment 2: Figures needs major attention specially the font size on axis and legends.

Response 2: We have accepted most of the suggestions and modified accordingly Figures 1, 2, 4 and 6, and Table 3. The profile plots are now moved to the new Supplemental Figure S2.

 Comment 3: Proper proofreading of the English language is necessary, as there are many errors.

Response 3: We’ve done our best to improve the English language.

Please, see also our point-by-point replies to your comments in the attached PDF file.

Reviewer 3 Report

Comments and Suggestions for Authors

In this manuscript authors created the fluorescence-fusion genes to directly analyze the generated fluorescence intensity image profiles in E. coli with some interested parameters. Overall authors did a good job, however there are some comments list as below:

1.     Since in this manuscript, authors focused on DnaA and Hu proteins only, both two proteins should be described for introduction. However, in the introduction part only presented DnaA but lacked basic information of Hu.

2.     In fig 1, the right column, the font is too small to be easily read/ distinguished, and the relative scales of X-axis in every panel is not same, but all panel were lined up, thus a miss-leading will be easily generated. Authors should change them to provide a much more accurate result.

3.     In fig S1, a typical E. coli growth curve should be plotted on a Semi-logarithmic coordinates system. It is better to change to clearly present the bacterial growth phases.

Reviewer 4 Report

Comments and Suggestions for Authors

The authors examine the distribution and diffusion rates of fluorescently tagged DnaA and HU in Escherichia coli. They find that that, as expected, the diffusion rate of DnaA-GFP is low compared to GFP but about 10 times the diffusion rate of HU. They further show that the concentrations of DnaA and HU vary considerably between individual cells, and propose that variable DnaA/DNA ratios result in variations in replication initiation time within the cell population. Although the research presented here is peripheral to my area of expertise, the manuscript is well written, the data are well documented, and the results are interesting.

Specific comments:

1. How representative are the images shown in Figure 1? It is difficult to assess what is meant by the statement “sometimes even being excluded from the nucleoid region” in reference to the distribution of the DnaA-GFP fusion in cells treated with chloramphenicol (see lines 107-108).

2. It is difficult to read the text in the graphs of the fluorescence intensities shown in the panels on the far right in Figure 1. While it is nice to have the graphs positioned next to the corresponding fluorescence microscopic images, it may make more sense to enlarge the graphs and put them in a separate panel below the micrographs.

3. The authors should describe the DnaA(L417P) mutant where they first introduce it on line 325. What is the nature of the DnaA mutant? Based on the results in the FRAP assay it would seem to be deficient in binding the DnaA-boxes on the chromosome, but this does not seem to be clearly stated anywhere in the manuscript.

Author Response

Comments and Suggestions for Authors

The authors examine the distribution and diffusion rates of fluorescently tagged DnaA and HU in Escherichia coli. They find that that, as expected, the diffusion rate of DnaA-GFP is low compared to GFP but about 10 times the diffusion rate of HU. They further show that the concentrations of DnaA and HU vary considerably between individual cells, and propose that variable DnaA/DNA ratios result in variations in replication initiation time within the cell population. Although the research presented here is peripheral to my area of expertise, the manuscript is well written, the data are well documented, and the results are interesting.

Specific comments:

Comment 1. How representative are the images shown in Figure 1? It is difficult to assess what is meant by the statement “sometimes even being excluded from the nucleoid region” in reference to the distribution of the DnaA-GFP fusion in cells treated with chloramphenicol (see lines 107-108).

Response 1: Thank you for raising this aspect specifically important for the variability studies. The images shown in Fig. 1 are well-representative. It is stated in the title of the figure: “The intracellular distribution of DnaA-GFP and HU-mCherry in a typical cell in different functional states.” In live cells, the profiles of HU and DnaA are, of course, age (size)-dependent, but their relative distribution is maintained as can be seen in various cells shown in Fig. 7, upper row. The population-wide (about 500 cells) colocalization is shown in Fig. 2 and summarized in Table 1.

The statement “sometimes even being excluded from the nucleoid region” is now changed to “the majority of DnaA fluorescence comes from the cytoplasm, even being essentially reduced in the nucleoid region of numerous cells” (lines 106-108).

Comment 2. It is difficult to read the text in the graphs of the fluorescence intensities shown in the panels on the far right in Figure 1. While it is nice to have the graphs positioned next to the corresponding fluorescence microscopic images, it may make more sense to enlarge the graphs and put them in a separate panel below the micrographs.

Response 2: We agree with this and other reviewer and, therefore, in the revised version the profile plots are omitted from this figure and moved to the Supplemental Material in enlarged size and fonts (new Fig. S2). 

Comment 3. The authors should describe the DnaA(L417P) mutant where they first introduce it on line 325. What is the nature of the DnaA mutant? Based on the results in the FRAP assay it would seem to be deficient in binding the DnaA-boxes on the chromosome, but this does not seem to be clearly stated anywhere in the manuscript.

Response 3: This mutant is first introduced on lines 294-295 of the originally submitted version specifying its malfunction: “In addition, plasmid-borne eGFP and DNA binding impaired mutant DnaA(L417P)-eGFP expressed in the parent MG1655 strain were used as free diffusing controls.”

Round 2

Reviewer 3 Report

Comments and Suggestions for Authors

In this revised version of manuscript, regarding to the authors response for three comments that had almost not done anything.

Point 1. Leaking basic information of Hu: authors only added one new sentence, it can not be accepted. Since authors used Hu to represent the coli DNA, and lots Hu background references can be easier searched out, such as: Hu protein is known as a DNA-binding protein, shares properties with histones, and is involved in nucleoid organization, DNA replication and transcription foci formation…. and so on. All above and more “common sense” of Hu protein should be addressed that could make much better for this manuscript introduction part.

Point 2. Authors only directly moved the “trouble” part to supplement part and omitted the “real revised work”. Authors have to change the relative scales of X-axis in every panel as the same scale to provide a much more accurate result and to avoid the miss-leading presentation by a firstly direct vision.

Point 3. Authors response to mention they used a semi-logarithmic coordinates system to plot the E. coli growth curve, however, in the fig S1, neither the title of y-axis nor figure legend can be found the relative description!

Round 3

Reviewer 3 Report

Comments and Suggestions for Authors

Here has no further questions.